# Private Graph All-Pairwise-Shortest-Path Distance Release With Improved Error Rate

**Chenglin Fan, Ping Li, Xiaoyun Li**

Cognitive Computing Lab
Baidu Research
10900 NE 8th St. Bellevue, WA 98004, USA
{chenglinfan2020, pingli98, lixiaoyun996}@gmail.com

## Abstract

Releasing all pairwise shortest path (APSP) distances among vertices on general graphs under weight Differential Privacy (DP) is known as a challenging task that has gained increasing interest recently. Previous work achieved DP with the maximal absolute error among all published pairwise distances bounded by $\tilde{O}(n)$ where $n$ is the number of nodes. Whether the approximation error can be reduced to sublinear in $n$ is still an interesting open problem. In this paper, we break the linear barrier on the distance approximation error in APSP release, by proposing an algorithm that releases a constructed synthetic graph privately. Computing all pairwise distances on the constructed graph only introduces $\tilde{O}(n^{1/2})$ error in answering all pairwise shortest path distances for fixed privacy parameter. Our method is based on a novel graph diameter (link length) augmentation via constructing "shortcuts" for the paths and the use of Laplace noise with non-zero mean. Numerical examples are also provided. Additionally, we also propose a DP algorithm with error rate $\tilde{O}(k)$, which improves the error of general graphs, when the graph has small feedback vertex set number $k = o(n^{1/2})$.

## 1 Introduction

In recent years, there has been a growing interest in private data analysis, from academic research to industry practice. Typically, many industrial machine learning applications consist of two steps: 1) collecting data from users; 2) training models using the collected user data. One key question is: throughout this process, how can we extract meaningful information from the data, without leaking sensitive data of each user? In other words, how can we prevent an adversarial attacker from inferring any user's data, given the public information that can be accessed? This has been the overarching question in a line of research that studies private algorithms under various settings. The concept of differential privacy (DP) (Blum et al., 2005; Chawla et al., 2005; Dwork, 2006) has been a popular approach for rigorously defining and resolving the problem of keeping useful information for model learning, while protecting privacy for each individual. In the traditional setting, databases $D$ and $D'$ are collections of data records and are considered to be neighboring if they are identical except for a single record (individual information). DP requires that the output of running a randomized algorithm on $D$ and $D'$ should have very close probability distributions. There are many types of databases, and in this work, we will specifically focus on the differential privacy of graphs.

**Weight private graphs.** In general, there are three types of private graph models, regarding the nodes, the edges and the edge weights, respectively. The node private model requires DP for two adjacent graphs differing in one node. In the edge private model, two neighboring graphs are defined such that they only differ by one edge, while sharing the same set of nodes.

36th Conference on Neural Information Processing Systems (NeurIPS 2022).

In this paper, we focus on the weight private graph model, which was first proposed by Sealfon (2016). In the weight private graph problem, the topology of the graph is public, which means that the adjacent graphs (databases) when considering DP have same nodes and edges, while this is not the case in other two DP models. Specifically, in the weight private model we consider two graphs as being adjacent if the difference between the sums of their edge weights is no more than one unit. One example application of this problem is where one tries to release the transportation volume between several places of interest with privacy constraints, where the roads are regarded as edges. In some sense, it might be hard to keep the graph structure private in practice, since one may easily obtain the map by modern tools like Google Earth. However, in some cases, information like traffic flows could also be sensitive and needs to be kept private. Releasing such private information is well suited for the setting of weight private graph model.

## 1.1 Open Problem, and Previous Results

In this paper, we study the problem of releasing all pairwise shortest-path (APSP) distances with weight privacy. Two standard strategies in DP are: 1) adding Laplace noise to each edge, 2) adding Laplace noise to the output distance matrix. As discussed in Sealfon (2016), the errors of these two strategies are roughly $O(\epsilon^{-1} n \log n)$ by adding noise to each edge and $O(\epsilon^{-1} n \sqrt{\log(1/\delta)})$ using DP composition theorems, respectively, where $n$ is the number of nodes in the graph and $(\epsilon, \delta)$ are the DP parameters. Here, the approximation error is measured by the largest absolute difference among all released pairwise distances and the ground truth. As we see, both of these errors are roughly $O(n)$, i.e., linear in $n$ when $\epsilon$ and $\delta$ are fixed. Whether we can achieve differential privacy with error sublinear in $n$ is still an open problem, as recognized by the online repository (https://differentialprivacy.org/open-problem-all-pairs/) focusing on the research of DP, quoted *"... Is this linear dependence on n inherent, or is it possible to release all-pairs distances with error sublinear in n ?"*

While this linear barrier seems to hold for general graphs, we can derive improved results on several special types of graph. Firstly, for a graph with weights bounded in $[0, M]$, Sealfon (2016) picked a subset $Z \in \mathbb{V}$ of vertices as the "$k$-covering set" to approximate the original graph. Each vertex $u$ can map to its closest vertex $z_u$ in the covering set. For any node pair $u, v$, their distance is approximated by the distance between $z_u$ and $z_v$ plus $O(kM)$ additional error. Then, the author proposed to use $O(|Z|^2)$ pairs of distances to approximate all $O(n^2)$ pairwise distances, which leads to $O(\sqrt{nM})$ approximation error for fixed privacy parameters eventually. Secondly, for grid graph with arbitrary positive weights, Fan and Li (2022) proposed to select an intermediate vertex set and divide the shortest path between any node pair into at most three parts depending on its traverse of the set. By connecting the nodes in the intermediate set and applying the standard Laplace mechanism, the authors constructed a DP algorithm with $\tilde{O}(n^{3/4})$ approximation error. Thirdly, for trees, Sealfon (2016) gave a recursive algorithm to release all-pairs distances with error $O(\log^{2.5} n)$, which was improved to $O(\log^{1.5} n \log^{1.5} h)$ by Fan and Li (2022), where $h$ is the depth of tree and can be as small as $O(\log n)$. The general idea was to build a collection $\mathbb{T}$ of subtrees of $\log n$ levels, where the subtrees in each level are disjoint to each other such that each edge appears in at most $O(\log n)$ trees, allowing us to only add $O(\log n)$ units of Laplace noise to each path in $\mathbb{T}$. This leads to $poly \log n / \epsilon$ approximation error on trees.

## 1.2 Our Results and Techniques

In this paper, we propose an algorithm that is able to surpass the linear $O(n)$ error of differentially privately releasing all pairwise distances on general weighted graphs. We make use of the improved and extended techniques from Fan and Li (2022) to general graphs to obtain $\tilde{O}(n^{1/2})$ approximation error. Moreover, when the graph exhibits some special property having small feedback vertex set number, we develop a new algorithm with further improved error bound.

**Breaking the linear error barrier for general graphs.** Let $G(\mathbb{V}, \mathbb{E}, w)$ denote an undirected graph, where $\mathbb{V}$ is the set of nodes with $|\mathbb{V}| = n$, $\mathbb{E}$ is the set of all edges, and $w$ is the set of corresponding weights. We use $w(e)$ to denote the weight of a specific edge $e \in \mathbb{E}$. The key challenge in this problem is that, we need to bound the error of all shortest paths instead of just one. Since the shortest paths could overlap with each other, the dependency among the shared edges (and the noises) hinders us from applying standard concentration results. To achieve sublinear error in DP all pairwise distance

release problem, our idea, intuitively, is to find "shortcuts" that will reduce the number of edges in long shortest paths (consisting of many edges). The algorithm proceeds as follows. Denote $d(u, v)$ as the true shortest path distance between node $u$ and $v$ w.r.t. graph $G(\mathbb{V}, \mathbb{E}, w)$. First, we randomly sample $n^{1/2}$ vertices from $\mathbb{V}$ to form a subset of vertices $\mathbb{V}_1 \subset \mathbb{V}$. We then create an edge set $\mathbb{E}_1 = \{(u, v) : u \in \mathbb{V}_1, v \in \mathbb{V}_1, u \neq v\}$ and assign each edge the weight $w_1(u, v) = d(u, v)$. Denote $\mathbb{E}_0 =: \{(u, v) \in \mathbb{E} : u \notin \mathbb{V}_1 \text{ or } v \notin \mathbb{V}_1\}$ as the edge set between nodes that are not both in $\mathbb{V}_1$. Next, we add noise to the edges in $\mathbb{E}_0$ and $\mathbb{E}_1$, respectively. For $e \in \mathbb{E}_0$, we add Laplace noise following $Lap(\mu_0, \sigma_0)$ where $\sigma_0 = O(1/\epsilon), \mu_0 = \sigma \log(n/\gamma)$; for $e \in \mathbb{E}_1$, the noise is from $Lap(\mu_1, \sigma_1)$ with $\sigma_1 := O(n^{1/2}\sqrt{\log(1/\delta)})/\epsilon)$ and $\mu_1 = \sigma_1 \log(n/\gamma)$. Note that, the mean of the Laplace noise is non-zero in our approach. We will show that this novel design guarantees that the APSP distances computed directly from the synthetic and noisy graph $G'$ are lower bounded by the true distances for all node pairs (w.h.p.), and the approximation error is at most $\tilde{O}(n^{1/2})$ (Theorem 3.6).

**Graph with small feedback vertex set number.** A feedback vertex set (FVS) of a graph, also called a loop cutset (Freuder, 1982), is a set of vertices whose removal leaves a graph without cycles, and the feedback vertex set number of a graph is the size of a smallest feedback vertex set. The feedback vertex set problem is NP-hard (Lewis, 1978) and the best known approximation algorithm on undirected graphs is by a factor of two (Becker and Geiger, 1996). Beyond the results derived for general weighted graphs, we also consider the problem for graphs with small feedback vertex set. We design a new algorithm that releases the distances privately with $\tilde{O}(k)$ error where $k$ is the feedback vertex set number, improving the error bound when $k = o(n^{1/2})$ compared with the result for general weighted graphs.

## 1.3 More Related Work

The topic of private graphs has attracted substantial research interests in the recent years, including edge privacy, node privacy and weight privacy (Hay et al., 2009; Rastogi et al., 2009; Gupta et al., 2010; Karwa et al., 2011; Gupta et al., 2012; Blocki et al., 2013; Kasiviswanathan et al., 2013; Bun et al., 2015; Sealfon, 2016; Ullman and Sealfon, 2019; Borgs et al., 2018; Arora and Upadhyay, 2019; Fan et al., 2022; Fan and Li, 2022). Our work focuses on the weight private problem (Sealfon, 2016) as described above. We generalize the previous work (Fan and Li, 2022) which focused on trees and grid graphs to general weighted graphs. We are also aware of two concurrent works (Chen et al., 2022; Ghazi et al., 2022). Both these two papers claim $\tilde{O}(n^{1/2})$ error upper bound same as our result, using a similar high-level idea of constructing "shortcuts". However, our approach is novel and different in that the Laplace noise we add has non-zero mean. As will be shown in the paper, this allows us to simply publish the constructed synthetic graph, upon which standard APSP computation leads to the desired error level. However, in both Ghazi et al. (2022) and Chen et al. (2022), the noise is centered and one has to use some specifically designed calculation (optimization) to obtain the estimated distances. In other words, only the estimated distances, but not the graph itself, can be published. Typically, releasing the graph (ours) is more difficult than releasing the distances only.

## 2 Background: Differential Privacy (DP) on Graph

Consider a connected graph $G = (\mathbb{V}, \mathbb{E}, w)$, with $w$ the collection of the weights of $\mathbb{E}$. We use $w(e)$ to denote the weight of an edge $e \in \mathbb{E}$. Denote $n = |\mathbb{V}|$, $m = |\mathbb{E}|$, with $n \leq m + 1$.

We first define the notion of neighboring graphs. In weight private model, since the nodes and edges of the graph $G$ are unchanged, we can simply use edge weights to represent the graph.

**Definition 2.1** (Neighboring). *Graph $G = (\mathbb{V}, \mathbb{E}, w)$ and $G' = (\mathbb{V}, \mathbb{E}, w')$ are called neighboring, noted as $G \sim G'$, if*

$$||w - w'||_1 := \sum_{e \in \mathbb{E}} |w(e) - w'(e)| \leq 1.$$

The Differential Privacy (DP) introduced by Dwork (2006) is defined below adapted to our problem.

**Definition 2.2** (Differential Privacy (Dwork, 2006)). *If for any two neighboring graphs $G = (\mathbb{V}, \mathbb{E}, w)$ and $G' = (\mathbb{V}, \mathbb{E}, w')$, a randomized algorithm $\mathbb{A}$, and a set of outcomes $O \subset Range(\mathbb{A})$,*

$$Pr[\mathbb{A}(G) \in O] \leq e^\epsilon Pr[\mathbb{A}(G') \in O] + \delta,$$

*we say algorithm $\mathbb{A}$ is $(\epsilon, \delta)$-differentially private.*

If $\delta = 0$, we say that the algorithm is $\epsilon$-DP. The parameter $\delta$ is usually interpreted as the probability allowed for bad cases where $\epsilon$-DP is violated. Intuitively, differential privacy requires that after changing the database by a little (the total weights in our case), the output should not be too different from that of the original database. Differential privacy may be achieved through the introduction of noise to the output. To attain general $(\epsilon, \delta)$-DP, we may add Gaussian noise to the output. To achieve $\epsilon$-DP, the noise added typically comes from the Laplace distribution. Smaller $\epsilon$ and $\delta$ indicate stronger privacy, which, however, usually sacrifices utility. Thus, one of the central topics in the differential privacy literature is to reduce the scale of noise added, while satisfying the privacy constraint. In this work, we focus on the popular approach to achieve DP by adding Laplace noises. The Laplace distribution with parameter $b$ has density function $f(x) = \frac{1}{2b} \exp(-|x|/b)$.

**Lemma 2.1** (Dwork (2006)). *For a function $f : \mathcal{G} \to \mathbb{R}$ with $\mathcal{G}$ the input space of graphs, define the sensitivity*

$$\triangle_f = \max_{G \sim G'} ||f(G) - f(G')||_1,$$

*where $G, G'$ are two neighboring graphs. Let $X$ be a random noise drawn from $Lap(0, \triangle_f/\epsilon)$. The Laplace mechanism outputs*

$$M_{f,\epsilon}(G) = f(G) + X,$$

*which achieves $\epsilon$-differential privacy.*

One important and attractive property of DP is that, different DP algorithms can be easily combined together, also with strict DP guarantee.

**Lemma 2.2** (Advanced Composition Theorem (Dwork et al., 2010)). *For any $\epsilon, \delta, \delta' \geq 0$, the adaptive composition of $k$ times $(\epsilon, \delta)$-differentially private mechanisms is $(\epsilon', k\delta + \delta')$-differentially private for*

$$\epsilon' = \sqrt{2k \log(1/\delta')} \cdot \epsilon + k \cdot \epsilon(e^\epsilon - 1),$$

*which is $O(\sqrt{k \log(1/\delta')} \cdot \epsilon)$ when $k \leq 1/\epsilon^2$. In particular, if $\epsilon' \in (0, 1), \delta, \delta' > 0$, the composition of $k$ times $(\epsilon, 0)$-differentially private mechanism is $(\epsilon', \delta')$-differentially private for*

$$\epsilon = \epsilon'/(\sqrt{8k \log(1/\delta')}).$$

Finally, we introduce a high probability upper bound on the maximum of Laplace random variables which will be used in the error analysis.

**Lemma 2.3.** *Consider $n$ i.i.d. random variables $Z_1, Z_2, ..., Z_n$ from $Lap(b)$. With probability at least $1 - \gamma$, $\forall 0 < \gamma < 1$, all $n$ Laplace random variables have magnitude bounded by $b \log(n/\gamma)$.*

*Proof.* For each $Z_i$, $P(|Z_i| \leq t) = 1 - \exp(-\frac{t}{b})$. Thus,

$$P[\max_{i=1,...,n} |Z_i| \leq t] = (1 - \exp(-\frac{t}{b}))^n.$$

Let $t = b \log(n/\gamma)$. With $n \geq 1$, the probability becomes $(1 - \frac{\gamma}{n})^n \geq 1 - \gamma$ as claimed. $\qquad\square$

## 3   Private Synthetic Graph Release for APSP Distance

We formally define the problem of interest and some basic concepts as below.

**Definition 3.1.** *(Approximate Distances Release) Given an graph $G(\mathbb{V}, \mathbb{E}, w)$, our task is to release all pairwise shortest path distances privately. Let $\hat{d}(\cdot, \cdot)$ be the output (approximate) distance function. Our object is to minimize the maximal absolute error over all pairs, namely, $\max\{|\hat{d}(u,v) - d(u,v)| : u, v \in \mathbb{V}\}$, with $\hat{d}(u,v), \forall u, v \in \mathbb{V}$ being DP to weight $w$. We call $|\hat{d}(u,v) - d(u,v)|$ the additive error of node pair $(u, v)$.*

**Definition 3.2** (Path). *A path $\tilde{P}_{v_1,v_L}$ between $v_1, v_L \in \mathbb{V}$ is defined as $\tilde{P}_{v_1,v_L} = \{(v_i, v_{i+1}), 1 \leq i \leq L-1\}$, the collection of edges between connected node sequence $v_1, v_2, ..., v_k$, with some $L \leq n$. A segment of path $\tilde{P}$ is defined as a **consecutive** sub-path, $\{(v_i, v_i + 1), s \leq i \leq t - 1\}$ for some $s, t$.*

**Definition 3.3** (Shortest path). *Let $S_{u,v}$ be the set of all paths between $u, v \in \mathbb{V}$. For a path $P \in S_{u,v}$ and weights $w$, denote $d(P, \mathcal{W}) = \sum_{e \in P} w(e)$. The shortest path of $u, v \in \mathbb{V}$ w.r.t. weights $w$ is*

$$P_{u,v} = \operatorname*{argmin}_{P \in S_{u,v}} d(P, w),$$

*and the shortest path distance is $d(P_{u,v}, w)$.*

**Definition 3.4** (Canonical shortest path). *For a given graph $G(\mathbb{V}, \mathbb{E}, w)$, let $\mathbb{V}_1 \subseteq \mathbb{V}$ be a subset of $\mathbb{V}$. Let $P_{x,y} = \{x, v_1, v_2, ..., v_L, y\}$ be the shortest path between $x, y \in \mathbb{V}$. Then a canonical shortest path $P_{x,y}^{\mathbb{V}_1}$ is defined by either of the following:*

1. *$P_{x,y}^{\mathbb{V}_1} \equiv P_{x,y}$, if $P_{x,y}$ contains at most one vertex in $\mathbb{V}_1$;*

2. *$P_{x,y}^{\mathbb{V}_1} = \{x, v_1, ..., p, q, ..., v_L, y\}$, if $p, q$ are the closest nodes in $P_{x,y}$ to $x, y$ respectively, and $p, q \in \mathbb{V}_1$.*

Intuitively, the canonical shortest path finds a shortcut by directly connecting two nodes in a set $\mathbb{V}_1$. This definition will be the key in our algorithm and construction. One important fact is that, if we connect each pair of nodes in $\mathbb{V}_1$ and assign the true pairwise distance between them as the edge weight (as in our main algorithm), then $d(P_{u,v}, w) \equiv d(P_{u,v}^{\mathbb{V}_1}, w)$ by definition.

## 3.1 Challenges and the New Algorithm

First, we revisit the prior approach to release all pairwise distances in the private weight model and the challenges. Since neighboring (total) weights differ by at most one unit in $l_1$ norm, the distance between any two nodes also changes by at most one unit. Releasing a single path can be done by computing the accurate shortest path distance between pairs of inputs and adding Laplace noise proportional to $1/\epsilon$, which is a trivial task in DP. However, releasing all distances privately is much more challenging, since the Laplace mechanism requires $Lap(n^2/\epsilon)$ noise (because of the $n^2$ queries), resulting in the $O(n)$ error eventually. As introduced in Section 1.1, in prior literature, there are two ways (e.g., Sealfon (2016)) to achieve this error level, either by adding noise to each edge or to the output distances. In this paper, we will focus on the first strategy.

The simple approach is as follows: 1) add $Lap(1/\epsilon)$ noise to each edge, i.e., $w' = w + Lap(1/\epsilon)$, to get graph $G' = (\mathbb{V}, \mathbb{E}, w')$; 2) report all pairwise distances on $G'$ (note that, the shortest paths on $G'$ found might be different from the true shortest paths in $G$). By the Laplace mechanism (Dwork, 2006), all the weights in graph $G$ become differentially private. By the post-processing property of DP, all the output pairwise distances are also DP. Note that there are $O(n^2)$ pairs of vertices, so the number of edges is bounded by $O(n^2)$. By Lemma 2.3, with probability $1 - \gamma$, all $O(n^2)$ Laplace random variables will have magnitude bounded by $(1/\epsilon) \log(n^2/\gamma)$, so the length of every path in the released synthetic graph is within $n \log(n/\gamma)/\epsilon$ additive error, thus roughly $O(n)$.

Statistically, the general problem can be described informally and approximately as: given a set of $n^2$ i.i.d. Laplace random variables, we want to bound the sum of the $n$ variables among $n^2$ size-$n$ subsets simultaneously. Since the graph topology can be arbitrary, it is not hard to find examples where getting a sublinear error seems impossible, and a straightforward additive bound exactly leads to the previous $O(n)$ error as mentioned above. In this section, we propose an algorithm that leverages the concept of canonical shortest paths (Definition 3.4), which finally leads to $\tilde{O}(n^{1/2})$ additive approximation error.

As summarized in Algorithm 1, for a general weighted graph $G(\mathbb{V}, \mathbb{E}, w)$, our proposed algorithm proceeds as follows:

1) Sample $n^{1/2}$ vertices from $\mathbb{V}$ uniformly to form set $\mathbb{V}_1$;
2) Create an edge set $\mathbb{E}_1 := \{(u, v) : u \in \mathbb{V}_1, v \in \mathbb{V}_1, u \neq v\}$. For each $e = (u, v) \in \mathbb{E}_1$, set $w(e) = d(u, v)$ as the true shortest path distance;
3) Add $Lap(\mu_1, \sigma_1)$ noise to each edge $e \in \mathbb{E}_1$, i.e., $w' = w + Lap(\mu_1, \sigma_1)$;
4) Add $Lap(\mu_0, \sigma_0)$ noise to each edge $e \in \mathbb{E}_0$, i.e., $w' = w + Lap(\mu_0, \sigma_0)$ where $\mathbb{E}_0 := \mathbb{E} \backslash \mathbb{E}_1$;
5) Obtain and release the merged graph $G' = (\mathbb{V}, \mathbb{E}' = \mathbb{E}_0 \cup \mathbb{E}_1, w')$. One can simply compute all pairwise distances on $G'$ as the estimation.

**Input:** General graph $G = (\mathbb{V}, \mathbb{E}, w)$, private parameter $\epsilon, \delta, \gamma$.
$\epsilon' = \epsilon/2$.
Sample $(n^{1/2})$ vertices from $\mathbb{V}$ uniformly and add them to $\mathbb{V}_1$.
Create an edge set $\mathbb{E}_1 := \{(u, v) : u \in \mathbb{V}_1, v \in \mathbb{V}_1, u \neq v\}$.
For each $e \in \mathbb{E}_1$, let $APSP(e)$ be the exact shortest path distance in $G$ between $(u, v)$ where
$\quad e = (u, v)$.
**for** *each edge $e \in \mathbb{E}_1$* **do**
$\quad\quad$ Let $\sigma_1 := (2\sqrt{2}n^{1/2}\sqrt{\log(1/\delta)})/\epsilon'$ and $\mu_1 := \sigma_1 \log(n/\gamma)$.
$\quad\quad$ Draw $X_e \sim Lap(\mu_1, \sigma_1)$.
$\quad\quad$ $w'(e) := APSP(e) + X_e$.
$\mathbb{E}_0 := \mathbb{E} \setminus \mathbb{E}_1$
**for** *each edge $e \in \mathbb{E}_0$* **do**
$\quad\quad$ Let $\sigma_0 := 1/\epsilon', \mu_0 := \sigma_0 \log(n^2/\gamma)$.
$\quad\quad$ Draw $X_e \sim Lap(\mu_0, \sigma_0)$.
$\quad\quad$ $w'(e) := w(e) + X_e$.
Compute all pairwise distances in graph $G' = (\mathbb{V}, \mathbb{E}' = \mathbb{E}_0 \cup \mathbb{E}_1, w')$.
**Output :** All pairwise distances of $G'$.

**Algorithm 1:** Private all pairwise shortest paths distance release.

**Comments.** Note that in Algorithm 1, the final estimated distances are calculated by using standard APSP distance computation on graph $G'$. One implication of this construction is that, beyond outputting the pairwise distances with privacy, we can in fact also publish the graph $G'(\mathbb{V}, \mathbb{E}', w')$ privately and allow the users to compute the private distances using standard algorithms by themselves. This is due to the positive mean of the Laplace noises. If we instead add zero-mean (centered) Laplace noises as in most standard approaches, then computing the APSP distances on $G'$ would not ensure the desired approximation error rate. We will provide more discussion on the role of the shifted noises at the end of this section.

Next, we provide theoretical analysis of our proposed algorithm. Firstly, recall that $\mathbb{V}_1$ contains $n^{1/2}$ uniformly sampled nodes from $\mathbb{V}$. Our analysis starts with the following fact.

**Lemma 3.1** (Sampling Intersection). *Suppose $U \subseteq \mathbb{V}$ is a fixed vertex set and $|U| = n^{1/2} \log(n^2/\gamma)$. With probability at least $1 - \gamma/n^2$, we have that $U \cap \mathbb{V}_1 \neq \emptyset$.*

*Proof.* Suppose we pick a random vertex $v$ from $\mathbb{V}$ each time, the probability that $v$ is not in $\mathbb{V}_1$ is $1 - 1/n^{1/2}$. Then the probability of interest is bounded by $Pr[U \cap \mathbb{V}_1 = \emptyset] \leq (1 - 1/n^{1/2})^{(\sqrt{n}\log(n^2/\gamma))} = \gamma/n^2$. $\qquad\square$

### 3.2 Privacy Analysis and Sublinear Error Bound

**Lemma 3.2.** *Algorithm 1 achieves $(\epsilon, \delta)$-DP.*

*Proof.* The privacy budget is divided into two parts:

(Part 1) The noise added to $\mathbb{E}_0$. It is obvious that for two neighboring inputs differ in the total weights of edges in $\mathbb{E}_0$ by 1, by adding Laplace noises according $Lap(\mu_0, 1/\epsilon')$, it achieve $(\epsilon', 0)$-DP. Note that adding a constant $\mu_0$ independent of the edge weights to the edges does not affect weight privacy.

(Part 2) The noise added to edges in $\mathbb{E}_1$. Similar arguments hold. There are at most $n^{1/2} \cdot n^{1/2} = n$ pairs in $\mathbb{E}_1$. Applying Laplace mechanism with composition theorem (Lemma 2.2), we know that adding noise following $Lap(\mu_1, \sigma_1)$ with $\sigma_1 = \sqrt{8n\log(1/\delta)}/\epsilon' = 2\sqrt{2}n^{1/2}\sqrt{\log(1/\delta)}/\epsilon'$ suffices to achieve $(\epsilon', \delta)$-DP. Applying simple DP composition theorem again proves the $(2\epsilon', \delta)$-DP. We conclude by noticing that $\epsilon' = \epsilon/2$. $\qquad\square$

We have the following fact that, any "long path" in $G$ would contain at least two nodes in $\mathbb{V}_1$ with high probability.

**Lemma 3.3.** *For a given path in $G(\mathbb{V}, \mathbb{E}, w)$ with number of edges larger than $2n^{1/2} \log(n^2/\gamma)$, it contains at least two vertices in $\mathbb{V}_1$ with probability $1 - 2\gamma/n^2$.*

*Proof.* Let us divide the path into three parts, such that two of them have at least $n^{1/2} \log(n^2/\gamma)$ edges. Based on Lemma 3.1, we have that each part above intersects with $\mathbb{V}_1$ with at least one vertex, each with probability at least $1 - \gamma/n^2$. As a result, by union bound, the whole path contains at least two nodes in $\mathbb{V}_1$ with probability $1 - 2\gamma/n^2$. $\square$

Recall $G' = (\mathbb{V}, \mathbb{E}' = \mathbb{E}_0 \cup \mathbb{E}_1, w')$ is the constructed synthetic graph with noisy weights. From now on we denote $w$ as the weight set containing the constructed shortcut weights, and define $G'_w = (\mathbb{V}, \mathbb{E}', w)$ as the graph with nodes $\mathbb{V}$, edges $\mathbb{E}'$ and weights $w$ (notice that it is different from $G' = (\mathbb{V}, \mathbb{E}', w')$). Consider those canonical shortest paths found by the true weights $w$ on $G'_w(\mathbb{V}, \mathbb{E}', w)$. We show that each canonical shortest path $P^{\mathbb{V}_1}_{u,v}$, $u, v \in \mathbb{V}$ in $G'_w$ has error $|d(P_{u,v}, w') - d(P_{u,v}, w)|$ bounded by $\tilde{O}(n^{1/2})$, where $d(P_{u,v}, w)$ is defined by Definition 3.3.

**Lemma 3.4.** *For any $u, v \in \mathbb{V}$, let $P^{\mathbb{V}_1}_{u,v}$ be the canonical shortest path (Definition 3.4) found by the true weights $w$. Then, it holds that $|d(P^{\mathbb{V}_1}_{u,v}, w') - d(P^{\mathbb{V}_1}_{u,v}, w)| = O(n^{1/2}\sqrt{\log(1/\delta)} \log^2(n/\gamma)/\epsilon)$ with probability $1 - 4\gamma$ for $\forall u, v \in \mathbb{V}$.*

*Proof.* Firstly, we have $|d(P^{\mathbb{V}_1}_{u,v}, w') - d(P^{\mathbb{V}_1}_{u,v}, w)| = |\sum_{e \in P^{\mathbb{V}_1}_{u,v}} (w'(e) - w(e))|$, and $w'(e) - w(e)$ is the corresponding Laplace noise added to edge $e$ in $P^{\mathbb{V}_1}_{u,v}$. The key observation in the proof is that, each canonical shortest path $P^{\mathbb{V}_1}_{u,v}$ can be divided into three parts $(u, ..., p), (p, q), (q, ..., v)$, where $p, q$ are the closest nodes in $\mathbb{V}_1$ to $u, v$, respectively, and the edge $(p, q)$ is the constructed shortcut provided by $\mathbb{V}_1$ and $\mathbb{E}_1$. Note that, $P^{\mathbb{V}_1}_{u,v}$ also might not contain a shortcut, but this does not matter in our analysis in the sequel. Denote $P_{u,v}$ as the shortest path between $u$ and $v$. Consider two cases:

- $|P_{u,v}| \leq 2n^{1/2} \log(n^2/\gamma)$. In this case, we know that the total length of $(u, ..., p)$ and $(q, ..., v)$ is bounded by $2n^{1/2} \log(n^2/\gamma)$. That is, $P^{\mathbb{V}_1}_{u,v}$ contains at most $2n^{1/2} \log(n^2/\gamma)$ edges from $\mathbb{E}_0$. By Laplace mechanism, with probability $1 - \gamma$, every edge in $\mathbb{E}_0$ leads to at most $O(\log(n/\gamma)/\epsilon)$ error. Thus, edges in $\mathbb{E}_0$ contribute at most $O(n^{1/2} \log^2(n/\gamma)/\epsilon)$ error. $P^{\mathbb{V}_1}_{u,v}$ also contains at most one shortcut edges from $\mathbb{V}_1$, which gives $O(n^{1/2} \log(n/\gamma)\sqrt{\log(1/\delta)}/\epsilon)$ error, with another probability $1 - \gamma$. Thus, with probability $1 - 2\gamma$, the error is bounded by $O(n^{1/2} \log^2(n/\gamma)/\epsilon)$.

- $|P_{u,v}| > 2n^{1/2} \log(n^2/\gamma)$. In this case, by Lemma 3.3 and union bound, we know that with probability $1 - \frac{2\gamma}{n^2} n^2 = 1 - 2\gamma$, $P^{\mathbb{V}_1}_{u,v}$ contains a shortcut $(p, q)$, and both the lengths of $(u, ..., p)$ and $(u, ..., q)$ are upper bounded by $n^{1/2} \log(n^2/\gamma)$. The remaining proof is similar to the arguments above. We have that with probability at least $1 - 4\gamma$, $|d(P^{\mathbb{V}_1}_{u,v}, w') - d(P^{\mathbb{V}_1}_{u,v}, w)|$ is bounded by $O(n^{1/2} \log^2(n/\gamma)/\epsilon)$.

In summary, we have shown that with probability at least $1 - 4\gamma$, $|d(P^{\mathbb{V}_1}_{u,v}, w') - d(P^{\mathbb{V}_1}_{u,v}, w)|$ is bounded by $O(n^{1/2} \log^2(n/\gamma)/\epsilon)$. The proof is complete. $\square$

Additionally, we have the following result stating that the estimated distance is no smaller than the true distance for all pairs of vertices, with high probability.

**Lemma 3.5.** *Let $P_{u,v}$ be the shortest path between $u, v \in \mathbb{V}$ on $G$, and $P'_{u,v}$ the shortest path found on $G'$. Then with probability $1 - 2\gamma$, $d(P'_{u,v}, w') \geq d(P_{u,v}, w)$ for all $u, v \in \mathbb{V}$.*

*Proof.* By adding each edge in $\mathbb{E}_1$ with noise according to $Lap(\mu_1, \sigma_1)$ where $\sigma_1 = 2\sqrt{2}n^{1/2}\sqrt{\log(1/\delta)}/\epsilon'$ and $\mu_1 = \sigma_1 \log(n/\gamma)$, with high probability $1 - \gamma$, every $Lap(\mu_1, \sigma_1)$ variable is greater or equal to $\mu_1 - \sigma_1 \log(n/\gamma) \geq 0$ with probability $1 - \gamma$ based on Lemma 2.3.

Similarly, for edges in $\mathbb{E}_0$, we add to each of them $Lap(\mu_0, \sigma_0)$ noise where $\sigma_0 = 1/\epsilon', \mu_0 = \sigma_0 \log(n^2/\gamma)$. Then with another high probability $1 - \gamma$, every variable according to $Lap(\mu_0, \sigma_0)$ is non-negative based on Lemma 2.3. Therefore, since with high probability the added noises on all the edges are positive, the noisy shortest path distance must be larger than the true distance, i.e., $d(P'_{u,v}, w')$. This proves the claim. $\square$

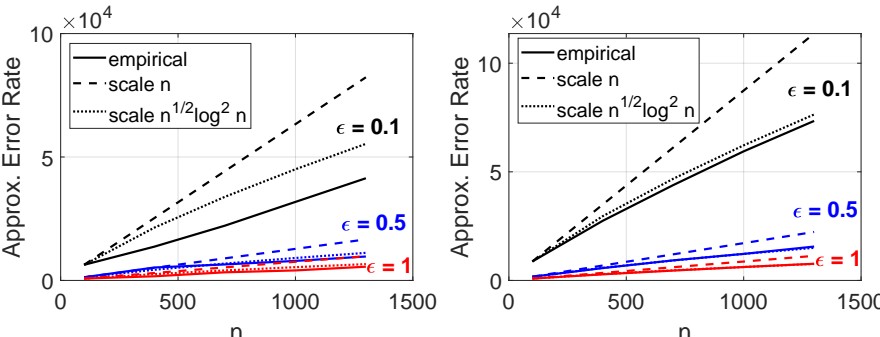

Figure 1: Empirical approx. error rate vs. graph size $n$ (solid curve), with predicted growth rate of $O(n)$ and $O(n^{1/2}\log^2 n)$ (dash and dotted curves). $\delta = 10$, $\gamma = 0.01$. Left: each edge weight is $Unif(2000, 3000)$. Right: edge weight from $Unif(10^4, 10^5)$. Averaged over 200 repetitions.

Now, we are in the position to present the main error guarantee of this work.

**Theorem 3.6.** *Let $G = (\mathbb{V}, \mathbb{E}, w)$ be general graph with $n$ vertices. For some $\epsilon, \delta > 0$, running Algorithm 1 publishes the graph $G'$ with $(\epsilon, \delta)$-differential privacy w.r.t. the weights $w$. Let $\hat{d}(u, v)$ be the estimated shortest path distance between $u, v \in \mathbb{V}$ computed on $G'$. Then, with probability $1 - 4\gamma$, we have $|\hat{d}(u, v) - d(u, v)| \leq O(\epsilon^{-1} n^{1/2} \log^2(n/\gamma)\sqrt{\log(1/\delta)})$ for all $u, v \in \mathbb{V}$.*

*Proof.* Let $P'_{u,v}$ be the shortest path between $u$ and $v$ in the synthetic graph $G'(\mathbb{V}, \mathbb{E}', w')$, then trivially, $d(P'_{u,v}, w') \leq d(P^{\mathbb{V}_1}_{u,v}, w')$. By Lemma 3.4, we know that $|d(P^{\mathbb{V}_1}_{u,v}, w') - d(P^{\mathbb{V}_1}_{u,v}, w)|$ is bounded by $O(n^{1/2}\sqrt{\log(1/\delta)}\log^2(n/\gamma)/\epsilon)$ with probability $1 - 4\gamma$, $\forall u, v \in \mathbb{V}$. In this event, we have $d(P'_{u,v}, w') \geq d(P_{u,v}, w)$ based on Lemma 3.5. Notice that, $d(P_{u,v}, w) \equiv d(P^{\mathbb{V}_1}_{u,v}, w)$ by definition. Therefore, with probability $1 - 4\gamma$,

$$|\hat{d}(u, v) - d(u, v)| = |d(P'_{u,v}, w') - d(P_{u,v}, w)| \underset{(a)}{\leq} |d(P^{\mathbb{V}_1}_{u,v}, w') - d(P^{\mathbb{V}_1}_{u,v}, w)|$$

$$\leq O(n^{1/2}\sqrt{\log(1/\delta)}\log^2(n/\gamma)/\epsilon), \quad (1)$$

which verifies the claim. $\qquad\square$

**Comments.** As we see, the positive mean of Laplace noises implies Lemma 3.5 which provides a key inequality in the proof of Theorem 3.6. This result in fact allows us to only consider the error on the positive side. On the contrary, if we use mean-zero Laplace noises, then we can no longer bound $|d(P'_{u,v}, w') - d(u, v)|$ by $|d(P^{\mathbb{V}_1}_{u,v}, w') - d(u, v)|$ as in Lemma 3.4, since it is possible that $d(P'_{u,v}, w') \leq d(P^{\mathbb{V}_1}_{u,v}, w') < d(u, v)$. Thus, if we simply add zero-mean noises, directly computing the APSP distances on the constructed graph $G'$ would not guarantee the desired error level.

### 3.3 Numerical Example

We present a numerical example on a multi-stage graph to justify the theory. We simulate a multi-stage graph, where each edge weight is generated from i.i.d. uniform distribution. The graph contains concatenated blocks, each with 1 start node, 1 end node, and 9 nodes in the middle connected to both the start node and the end node. The end node then becomes the start node of the next block. We consider multi-stage graphs here because in some sense it is one of the "hardest" cases for private APSP computation, since many shortest paths contain $O(n)$ edges. In Figure 1, we present the empirical largest absolute error over all pairwise distances against the size of graph $n$. For the theoretical bounds, we start with the first empirical error (black) associated with $n = 100$, and increase at the rate of $n$ (dash) and $n^{1/2}\log^2 n$ (dotted), respectively. We validate that the empirical error grows with rate slower than $n^{1/2}\log^2 n$, as given by our theory, for all $\epsilon$ values. Note that the error bound becomes tighter with large edge weights (right) than with small weights (left). When the edge weights follow $Unif(10^4, 10^5)$ which is much larger than the noise magnitude, the shortest path $P'_{u,v}$ found on $G'$ would be the same as $P_{u,v}$ (or $P^{\mathbb{V}_1}_{u,v}$) in most cases. Thus, the LHS and the RHS of (a) in (1) would be close, making the bound tighter.

# 4 Graph with Small Feedback Vertex Set

**Input:** General graph $G = (\mathbb{V}, \mathbb{E}, w)$, private parameter $\epsilon, \delta, \epsilon' = \epsilon/3$.
Compute the FVS $S$ using the 2-approx. algorithm in Becker and Geiger (1996).     // Step 1)
Let $G'$ be the induced subgraph of vertex set $\mathbb{V} \setminus S$ in $G$. Let $k := |S|$.
Compute the differentially private all pairwise shortest path distances $\hat{d}(u, v)$ on trees/forest $G'$
 using Algorithm 1 in Sealfon (2016) with privacy parameter $\epsilon'$.             // Step 2)
Compute $d(u, v)$ for $u, v \in S$, where $d(u, v)$ is the exact shortest path distance in $G$.  // Step 3)
**for** *each $u \in S, v \in S$* **do**
  Let $\sigma_1 := (2\sqrt{2}k\sqrt{\log(1/\delta)})/\epsilon'$.
  Draw $X_e \sim Lap(0, \sigma_1)$.
  $\hat{d}(u, v) := d(u, v) + X_e$.
**for** *each vertex $u \in S$* **do**
  **for** *each vertex $v \notin S$ adjacent to $u$* **do**
  // Step 4): To deal with the edges between $S$ and $\mathbb{V} \setminus S$
  Let $\sigma_0 := 1/\epsilon'$.
  Draw $X_e \sim Lap(0, \sigma_0)$.
  $w'(u, v) := w(u, v) + X_e$.
  $\hat{d}(u, v) := w'(u, v)$.
**for** *each vertex $u$ in $\mathbb{V} \setminus S$* **do**
  **for** *each vertex $v$ in $S$* **do**
    // Step 4): To deal with the case that only the last vertex of shortest path is in $S$
    $\hat{d}(u, v) := \min_{p \notin S}\{\hat{d}(u, p) + w'(p, v)\}$.
**for** *each vertex $u$ in $\mathbb{V} \setminus S$* **do**
  **for** *each vertex $v$ in $S$* **do**
    // Step 5): To deal with the case that shortest path passes a vertex $p$ in $S$
    $\hat{d}(u, v) := \min_{p \in S}\{\hat{d}(u, p) + \hat{d}(p, v)\}$.
**for** *each pair $u, v \in \mathbb{V} \setminus S$* **do**
  // Step 6): To deal with the case that shortest path passes some vertex $u$ in $S$
  $\hat{d}(u, v) := \min_{p \in S}\{\hat{d}(u, p) + \hat{d}(p, v)\}$.
**Output :** All pairwise distances $\hat{d}(u, v)$.

**Algorithm 2:** Differentially private all pairwise shortest path distances release for graph with small feedback vertex number.

In previous section, we have investigated the problem of DP all pairwise shortest distance problem on general weighted graphs. In this section, we consider a more special type of graph with small feedback vertex set (FVS), which is a common concept in graph analysis, e.g., (Kratsch and Schweitzer, 2010; Papadopoulos and Tzimas, 2020). We will design a new algorithm based on FVS computation that privately releases all pairwise distances. We will show that this method improves the error on general graphs when the FVS number is small. To begin with, the definition of FVS is given as below.

**Definition 4.1** (Feedback Vertex Set (Becker and Geiger, 1996))**.** *Let $G = (\mathbb{V}, \mathbb{E})$ be an undirected graph. A set $X \subseteq \mathbb{V}$ is called a feedback vertex set (FVS) if $G \setminus X$ is a forest, where $G \setminus X$ is the induced graph by $\mathbb{V} \setminus X$. The feedback vertex set number is the minimal cardinality over all the possible feedback sets.*

In words, a feedback vertex set (FVS) of a graph is a subset of vertices $X$ such that after removing these nodes from the graph, the subgraph induced by $\mathbb{V} \setminus X$ contains no cycles (i.e., is a forest). The FVS number is the smallest size of all the FVS's.

Next, we propose a differentially private algorithm that releases the APSP distances of graphs based on the feedback vertex set computation. The steps of Algorithm 2 are summarized below:

1) We compute a feedback vertex set $S$ of $G$ by using the 2-approximation algorithm in Becker and Geiger (1996), with $|S| = k$; the induced subgraph $G'$ of vertex set $\mathbb{V} \setminus S$ is a forest;

2) We use Algorithm 1 in Sealfon (2016) to obtain the private all pairwise shortest path distances in $G'$. For any pair $(u, v)$ in $G$ whose shortest path does not pass any vertex in $S$, $\hat{d}(u, v)$ on $G'$ is already a good estimation for $d(u, v)$;

3) For the $k^2$ (recall $k = |S|$) pairwise distances of the node pairs in $S$, we can compute the APSP distances directly and add $Lap(0, \sigma_1)$ to achieve DP. We still use the $\hat{d}(u, v)$ to represent them;

4) For each edge $(u, v)$ with $u \in S, v \in \mathbb{V} \setminus S$, we add noise according $Lap(0, \sigma_0 = 1/\epsilon')$ to $w(u, v)$. For a pair in $\{(u, v) | u \in \mathbb{V} \setminus S, v \in S\}$, if the shortest path between $u$ and $v$ does not pass other vertex in $S$ except $v$, then there exist some neighbor $p$ of $v$ such that $d_G(u, v) = d_{G'}(u, p) + w(p, v)$ based on Lemma A.1, where $d_G(u, v), d_{G'}(u, v)$ represent the shortest path distance between $u$ and $v$ in $G$ and $G'$ respectively. Hence we use $\hat{d}(u, v) := \min_{p \notin S} \{\hat{d}(u, p) + w'(p, v)\}$ to estimate $d(u, v)$;

5) For a pair in $\{(u, v) : u \in \mathbb{V} \setminus S, v \in S\}$, if the shortest path between $u$ and $v$ passes some vertex $p \in S$ except $v$, then there exist $p \in S, p' \notin S$ such that $d_G(u, v) = d_G(u, p) + d_G(p, v)$, and the shortest path between $u$ and $p$ does not pass any vertex in $S$ except $p$ based on Lemma A.1. We obtained $\hat{d}(u, p)$ in step 4). We then use $\hat{d}(u, v) := \min_{p \in S} \{\hat{d}(u, p) + \hat{d}(p, v)\}$ to estimate $d(u, v)$;

6) For pairs in $\{(u, v) \in (\mathbb{V} \setminus S)^2\}$, if the shortest path between $u$ and $v$ passes some vertex $p$ in $S$, then $d_G(u, v) = d_G(u, p) + d_G(p, v)$. We can use $\hat{d}(u, v) := \min_{p \in S} \{\hat{d}(u, p) + \hat{d}(p, v)\}$ to estimate $d(u, v)$, also $\hat{d}(u, p), \hat{d}(p, v)$ had been computed in 4), 5).

The auxiliary lemma mentioned in step 4) and 5) is given Lemma A.1 in the Appendix.

## 4.1 Privacy and Error Analysis

**Lemma 4.1.** *Algorithm 2 achieves $(\epsilon, \delta)$-DP.*

The proof is placed in the Appendix. The error bound of Algorithm 2 is given as below.

**Theorem 4.2.** *Let $G = (\mathbb{V}, \mathbb{E}, w)$ be a general graph with $n$ vertices. For some $\epsilon, \delta > 0$, running Algorithm 2 publishes all APSP distances that is $(\epsilon, \delta)$-differentially private w.r.t. the weights $w$. With probability $1 - 3\gamma$, the additive error is bounded by $O(\epsilon^{-1} k \log(k/\gamma) \sqrt{\log(1/\delta)} + \epsilon^{-1} (\log^{2.5} n) \log(1/\gamma))$.*

**Comments.** We see that when $k = |S| = o(n^{1/2})$, the error in Theorem 4.2 is smaller than the $O(n^{1/2} \log^2 n)$ error in Theorem 3.6 for the general weighted graphs. Therefore, our result implies improved error bounds when the graph has small feedback vertex set such that $k = o(n^{1/2})$. Further, if $k = o(\log n)$, i.e., the graph is "almost" a forest, then the first term vanishes and the error reduces to $\tilde{O}(\log^{2.5} n)$ for trees (Sealfon, 2016).

## 5 Conclusion

In literature, to achieve $(\epsilon, \delta)$-differential privacy (DP) in general weight private graphs, releasing all pairwise shortest path (APSP) distances incurs maximal approximation error $\tilde{O}(n)$, linear in the number of vertices. Recently, Fan and Li (2022) studied this problem on grid graphs and trees, but the linear error barrier for general weighted graphs is still an open problem (Sealfon, 2020). In this work, we propose a differentially private algorithm to release a carefully constructed graph, and computing the APSP distance on the synthetic graph achieves $\tilde{O}(n^{1/2})$ error sublinear in $n$, thus answering the open question. With improved and extended techniques from Fan and Li (2022), the idea of our approach is to augment the diameter of graph by a set of shortcuts along with adding shifted Laplace noise. Moreover, for graphs with small feedback vertex set number $k$, we also propose a DP algorithm to answer all pairwise shortest path distances each with error $\tilde{O}(k)$. This improves the result for general graphs when $k$, the feedback vertex set number, is small.

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
