# A  Missing Proofs in Section 4

**Lemma A.1.** *For a given graph $G(\mathbb{V}, \mathbb{E}, w)$, let $G'$ be the induced subgraph of $G$ with vertex set $\mathbb{V} \setminus S$. For any pair $\{(u,v)|u \in \mathbb{V} \setminus S, v \in S\}$: 1) if the shortest path between $u$ and $v$ does not pass other nodes in $S$, then there exist a neighbor $p$ of $v$ such that $d_G(u,v) = d_{G'}(u,p) + w(p,v)$; 2) if the shortest path $P_{u,v}$ between $u$ and $v$ does pass other nodes in $S$ except $v$, then there exist vertices $p \in S, p' \notin S$ such that $d_G(u,v) = d_G(u,p) + d_G(p,v)$ and $d_G(u,p) = d_{G'}(u,p') + w(p',p)$.*

*Proof.* 1) Note that the shortest path between $u$ and $v$ does not pass other nodes in $S$ except $v$. Let the second last vertex from $u$ to $v$ be $p$, with $p \notin S$. Then we have $d_G(u,p) = d_{G'}(u,p)$, thus $d_G(u,v) = d_G(u,p) + w(p,v) = d_{G'}(u,p) + w(p,v)$. 2) Let $p$ be the first vertex in $S$ that $P_{u,v}$ passes through, and denote $p'$ as the vertex just before $p$ in path $P_{u,v}$. We have that $d_G(u,p') = d_{G'}(u,p')$. Then we know that $d_G(u,v) = d_G(u,p) + d_G(p,v)$ and $d_G(u,p) = d_{G'}(u,p') + w(p',p)$. $\square$

## A.1  Proof of Lemma 4.1

*Proof.* The privacy budget $\epsilon$ is divided into three parts:

(Part 1) For the distances between all pairs in $\mathbb{V} \setminus S$, our method achieves $(\epsilon', 0)$-DP by the result on trees from (Sealfon, 2016).

(Part 2) For the $k^2$ distances of all node pairs in $S$, by adding to each edge weights i.i.d. $Lap(0, \sigma_1)$ noises with $\sigma_1 = \sqrt{8n \log(1/\delta)}/\epsilon' = 2\sqrt{2}n^{1/2}\sqrt{\log(1/\delta)}/\epsilon'$, we can achieve $(\epsilon', \delta)$-DP according to Lemma 2.2.

(Part 3) For each edge $(u,v)$ with $u \in \mathbb{V} \setminus S, v \in S$, obviously by Laplace mechanism $w'(u,v) := w(u,v) + Lap(1/\epsilon')$, we achieve $(\epsilon', 0)$-DP to release all the pairwise distances.

Composing three privacy budges up and using simple composition theorem of DP, we show that Algorithm 2 achieves $(\epsilon, \delta)$-DP. $\square$

## A.2  Proof of Theorem 4.2

**Lemma A.2** (All pairwise distance on trees (Sealfon, 2016))**.** *For a tree $T$ with non-negative edge weights $w$ and $\epsilon > 0$, there is an $\epsilon$-differentially private algorithm that releases APSP distances such that with probability $1 - \gamma$, all released distances have approximation error bounded $O(\log^{2.5} n \log(1/\gamma)/\epsilon)$.*

*Proof.* (of Theorem 4.2) For $u \in S, v \in S$, by adding $d(u,v)$ with $Lap(0, \sigma_1)$ noise where $\sigma_1 = 2\sqrt{2}n^{1/2}\sqrt{\log(1/\delta)}/\epsilon'$, with probability $1 - \gamma$, $|\hat{d}(u,v) - d(u,v)| = O(k\sqrt{\log(1/\delta)}\log(k/\gamma))$ $\forall u,v \in S$. Based on Lemma A.2, we have that with probability $1 - \gamma$, all released distances in $G'$ have approximation error $O(\log^{2.5} n \log(1/\gamma)/\epsilon)$. For those edges $(u,v)$ with $u \in \mathbb{V} \setminus S, v \in S$, we add to each $w(u,v)$ a Laplace noise according to $Lap(0, \sigma_0)$ with $\sigma_0 = 1/\epsilon'$. Thus, with another probability $1 - \gamma$, $|w'(u,v) - w(u,v)| \le O(\log(n/\gamma)/\epsilon)$ $\forall \{u \in \mathbb{V} \setminus S, v \in S\}$. Union bound implies that, with probability $1 - 3\gamma$, the total error is bounded by $O(\epsilon^{-1}k\log(k/\gamma)\sqrt{\log(1/\delta)}) + O(\epsilon^{-1}(\log^{2.5} n)\log(1/\gamma))$ as claimed. $\square$