# OpenReview forum: "Private Graph All-Pairwise-Shortest-Path Distance Release with Improved Error Rate"
_NeurIPS.cc/2022/Conference — NeurIPS 2022 Accept_

### Official Review · Reviewer_a4xF · 2022-06-17

**Rating:** 7
**Confidence:** 4
**Soundness:** 3 good
**Presentation:** 3 good
**Contribution:** 4 excellent

**Summary:**

This paper studies the problem of computing differentially private (DP) estimates of all-pairs shortest path distances (APSP). In this setting, we are given a weighted graph $G = (V, E, w)$ and the goal is to output the shortest path distances between all pairs of vertices; the error is defined as the maximum error among all pairs of vertices. DP constraint here is on the *weights*. Specifically, it is assume that the underlying graph $(V, E)$ is public and two weights $w, w'$ are neighbors if $\|w - w'\|_1 \leq 1$. DP then constrains that the output distribution on two such neighboring weights should not change too much. The changes are parameterized by two parameters $\epsilon, \delta$.

Sealfon (PODS 2016) was the first to study the question and gave an $\epsilon$-DP algorithm with error $O(n)$ for general graphs and $O(\log^{2.5} n)$ on trees. The main result of this paper is an $(\epsilon, \delta)$-DP algorithm with improved error of $\tilde{O}(\sqrt{n})$ in general graphs. The algorithm works roughly as follows.
- First, add "shifted" non-negative version of Laplace noise to each edge. Note that here we may select Laplace noise to have std $O(1/\epsilon)$.
- Second, randomly select $\sqrt{n}$ "shortcut" vertices, compute shortest paths (in $G$) between all pairs of shortcuts, add Laplace noise to these paths and then add these shortcut pairs as edges. Note that in this case there are only $n$ pairs and therefore the STD of the noise required is only $\tilde{O}(\sqrt{n} / \epsilon)$ due to advanced composition theorem.
- Compute the APSP in this new noised and "shortcut-augmented" graph.

The crux of the analysis is based on the fact that w.h.p. any shortest path from between nodes $u$ to $v$ can be decomposed into three parts: a "short" path $u$ to a shortcut, a path from a shortcut to another shortcut, and a "short" path from the second shortcut to $v$. "Short" here means that the path consists of at most $\tilde{O}(\sqrt{n})$ edges. Since we add Laplace noise with STD $O(1/\epsilon)$ to each edge, this roughly means that the error in the first and last part is $\tilde{O}(\sqrt{n}/\epsilon)$. On the other hand, we add Laplace noise with STD $\tilde{O}(\sqrt{n}/\epsilon)$ to APSP of shortcuts, so the error in the second part is also $\tilde{O}(\sqrt{n}/\epsilon)$. In total, we have that the error is $\tilde{O}(\sqrt{n}/\epsilon)$.

Finally, the authors extend the algorithm to graphs that have small feedback vertex set (FVS), which is the set of vertices whose removal makes the graph a forest. The algorithm proceeds by using vertices in FVS as shortcut, and then use Sealfon's algorithm for trees for the vertices outside. In total, this results in an error that is nearly-linear in the size of the FVS.

**Questions:**

- Can you please clarify the similarities / differences between this work and the other two concurrent works?

### Comments for Authors (Non-Questions)

- DP definition (Definition 2.2): the quantifier for $O$ is unclear. Please consider rephrasing.
- I find the way noise random variables are defined in the algorithms to be strange. Currently, they are written as $X_e := Lap(\mu, \sigma)$. This is a bit strange; it almost looks like you define $X_e$ to be the distribution. Please consider rephrasing / changing the notation.
- The notion for Laplace noise is also confusing: sometimes you write $Lap(\mu, \sigma)$ but sometimes only $Lap(\sigma)$. Please consider fixing this.
- Line 165: I think the noise $Lap(n^2/\epsilon)$ should be changed to $Lap(O(n\sqrt{\log(1/\delta)}/\epsilon))$ due to advanced composition.
- Line 177-180: "To our best knowledge, this problem has no solution sublinear in $n$ in literature". Actually it is pretty obvious to see that getting sublinear in $n$ is impossible. Let's consider the following example in the DP APSP setting: consider a graph where there are $n/2$ layers, each node connected to the consecutive layers, and let the weight of each edge be equal to some large number $W$. In this case, the original distance between the leftmost and rightmost nodes are $W(n/2 - 1)$, i.e. we pay $W$ to move to the next layer. Now, if you add Laplace noise to each entry, then we can continue moving from left to right while picking the lighter of the two edges. It can be seen that the expectation of the weight of such an edge is $W - \Omega_\epsilon(1)$. In total, this means that we get an expected error of $\Omega_\epsilon(n)$.
- Section 3.3: I find the graph description to be vague. (E.g. the graph is "multi-stage" but currently only one stage is described.) Please add more details on how exactly are the graph generated.

**Limitations:**

No ethical concerns about this work as far as I can tell.



**Strengths And Weaknesses:**

## Strengths

- APSP is one of the most important problems in algorithmic graph theory and is used as subroutine for many tasks. Therefore, it is likely that DP algorithm for APSP can be highly impactful. (In fact, as stated in the paper, this is a moderately well-known open problem in DP community.)
- The algorithms presented are simple and practical.

## Weaknesses

I do not see major weaknesses. A couple minor points are:
- There are two concurrent papers that seem to be achieving a similar result (in fact one of them seems to be getting better by some polylogarithmic factor). The current paper fails to discuss this, so it is unclear what the relationships / differences / similarities between the papers are.
- The writing can be improved (see "Comments for Authors" below).

## Evaluation

Given that the paper makes progresses on a fundamental problem in DP graph analysis and that the algorithm is simple & practical, I support acceptance.

---

> ### Author Response · Authors · 2022-08-02
> **Response to Reviewer a4xF**
>
> Thanks for your detailed summary of our contributions and support of our paper.
>
> W1 \& Q1: Can you please clarify the similarities / differences between this work and the other two concurrent works?
>
> Reply: While the NeurIPS conference website says "papers appearing less than two months before the submission deadline are not expected to be compared", we acknowledged and discussed the two concurrent works in Section A of the appendix. We are happy to mention these two papers in the main paper. We are glad to see more research interests and results on this problem and have descent respect to every try in this topic.
>
> Differences: Our approach is different from these two concurrent papers, in that the Laplace noise we add has non-zero mean. This allows us to simply publish the constructed synthetic graph, upon which standard APSP computation leads to the desired error level. In contrast, in both [Ghazi et al. 2022] and [Chen et al. 2022], the noise is centered and one has to use some specifically designed calculation (optimization) to obtain the estimated distances. In other words, only the estimated distances, but not the graph itself, can be published. Typically, releasing the graph (ours) is more difficult than releasing the distances only (theirs).
>
> Theory: Our paper and the two concurrent papers all implemented a similar idea of constructing shortcuts in some way, and proved the $\tilde O(\sqrt{n})$ error in the most general case. Ghazi et al. obtained a lower bound of $\Omega(n^{1/6})$, and Chen et al. showed an error slightly better than $O(n^{1/2})$ under the constraint of bounded weights. In our paper, we show that the error can be $O(k)$ for graphs with small vertex set number $k$.

---

> > ### Comment · Reviewer_a4xF · 2022-08-03
> > **Thank you**
> >
> > I'd like to thank the authors for the reply. The relationships to concurrent works are more clear to me now. I'd also like to apologize for not noticing Appendix A earlier (which led to the question).

---

> > > ### Author Response · Authors · 2022-08-03
> > > **Thanks for the reply**
> > >
> > > Thank you for the reply. We are glad that our response answers your question. Although these two papers were posted online within one month before NeurIPS deadline, we did not hesitate to cite them since we felt that as a researcher, we should stay truthful and respectful to every effort in the community. In fact, we are very happy that you mentioned these two works without seeing our discussion in the appendix, which means this problem/direction is getting more and more interests from the community. Again, we sincerely appreciate your valuable feedback, support and discussion.

---

### Official Review · Reviewer_USDQ · 2022-07-06

**Rating:** 6
**Confidence:** 4
**Soundness:** 4 excellent
**Presentation:** 2 fair
**Contribution:** 3 good

**Summary:**

This paper investigates how to release all-pairs shortest paths under edge
differential privacy. The paper achieves an sublinear (\sqrt{n}) error for this
problem using a vertex sampling scheme and adding Laplace noise (plus a positive
constant) to the edge weights. They solve the more general problem of
releasing a graph privately whose shortest paths match those of the input graph
up to the stated bounds.

Furthermore, they develop a second algorithm which computes all-pairs shortest
paths in a graph with bounded feedback vertex set size. This assumption allows
them to leverage the tree-like properties of these graphs which enables better
error when this number is less than \sqrt{n}.

**Questions:**

  Introduction: what are examples of graphs with small feedback vertex set?
  Does the second algorithm have practical interest?

  Related work: what are all of these papers and why are they somewhat relevant
  to this setting?

  Section 3.2: The graph $G_w' = (V, E', w)$ doesn't make sense because $w$ is
  defined on just the original set $E$ of edges, whereas $E'$ contains more
  edges. The explanations after Lemmas 3.3-3.5 are not providing much intuition
  or a clear explanation.

  Section 4: How does the FBS algorithm work on a high level? What do steps 1-5
  mean? What is the point of Lemma 4.1?

**Limitations:**

There are no real limitations to this work.

**Strengths And Weaknesses:**

+ The algorithm achieves sublinear error which is often not possible for DP algorithms.

+ The algorithm is simple and would be straightforward to implement. The
  computational overhead is not too high and only involves releasing a graph
  with O(n) more edges than the input graph.

+ Their improved algorithm for graphs with bounded feedback vertex set is a good
  generalization of the core ideas to a large class of graphs.

- The writing of the paper is lacking. The section organization, the notation,
  and the explanations, particularly in the algorithm sections, can be confusing
  or non-existent.
  The paper could also use general style and language improvements.

- Their experiments are not very strong, providing just an example on a uniformly
  random graph (which quite different from real-world graphs).

---

> ### Author Response · Authors · 2022-08-02
> **Response to Reviewer USDQ**
>
> Thanks for your valuable feedback on our paper.
>
>
> Q1: Introduction: what are examples of graphs with small feedback vertex set? Does the second algorithm have practical interest?
>
> Reply:  In practice, there are common applications where the graph has small feedback vertex number. One particular example is the transportation routing maps like the New York City MTA map and the US railway map. For these graphs, the number of feedback vertices is smaller than the number of interchange stops/cross stations, which is usually small compared to the graph size. After removing these nodes, the remaining graph becomes a forest. Thus, the error can be very small according to our theory in Section 4.
>
>
> Q2: Related work: what are all of these papers and why are they somewhat relevant to this setting?
>
> Reply: The references in Section 1.3 are all publications on graph privacy. They can be classified into three categories: node privacy, edge privacy and weight privacy. The node privacy focuses on privacy of node removal, and the edge privacy considers privacy w.r.t. edge removal. The two most related papers are [Seafon 2016, Fan and Li 2022], which considered weight privacy as studied in our work. We did not include detailed description due to the limited space of the submission. We will add more description in the revision for clarity.
>
>
>
> Q3: Section 3.2: The graph
>  doesn't make sense because  is defined on just the original set  of edges, whereas  contains more edges. The explanations after Lemmas 3.3-3.5 are not providing much intuition or a clear explanation.
>
> Reply: When we define $G_w'$ at line 219, $w$ has already contained the new weights of shortcuts in $E_1$ (line 187 in the algorithm, $w(e)=d(u,v)$ implies adding the edge weight $e$ to $w$). We will clarify this notation for clarity. Thank you.
>
> Lemma 3.3 says that with high probability, every shortest path in the original graph can be split into 2 subpaths not containing nodes in $V_1$ and one constructed shorcut.
>
> Lemma 3.4 bounds the error of this path by adding up the error of the three parts.
>
> Lemma 3.5 says that the noisy shortest path distance is no smaller than the true distance with high probability.
>
> These three lemmas together prove Theorem 3.6.
>
>
> Q4: Section 4: How does the FVS algorithm work on a high level? What do steps 1-5 mean? What is the point of Lemma 4.1?
>
> Reply:  The high level idea of FVS algorithm: The vertices in $G$ can be divided into two sets $S$ and $V\setminus S$, where $S$ is a feedback vertex set of $G$. Then $G(V\setminus S)$ is the induced subgraph (a forest) of vertex set $V\setminus S$ in $G$. (i) For the shortest path with all nodes in $G(V\setminus S)$, we can use the existing method in literature to deal with APSP in tree $G(V\setminus S)$. (ii) For the pairs with both vertices in $S$, we output the noised distances (exact distance plus noise) directly.
> (iii) For the shortest paths between $s,t$ in $G$ passing some vertices in $S$, we can do a decomposition of $d(s,t)$ into several segments and sum up the error, similar to the idea of the general graph algorithm. Lemma 4.1 presents the precise statement of this decomposition.
>
> The role of each step is explained as follows.
>
> Step 1: find the feedback vertex set $S$;
>
> Step 2: private distance computation of case (i) above;
>
> Step 3: private distance computation of case (ii) above;
>
> Step 4-6: private distance computation of case (iii) above;
>
> We will add some explanations to make the idea more clear. Thanks also for suggestions on improving the presentation of the paper.

---

### Official Review · Reviewer_vGTs · 2022-07-10

**Rating:** 6
**Confidence:** 3
**Soundness:** 4 excellent
**Presentation:** 3 good
**Contribution:** 3 good

**Summary:**

Overview of Problem:
This paper studies the problem of all-pairs shortest path distances (APSP) with differential privacy. In this setting, we are given an weighted graph, where the set of vertices and edges are known (and public), but the weights should be kept private. Since we cannot reveal the edge weights, the goal is to approximate every shortest path approximately up to small error. The model of privacy is where two weighted graphs are adjacent if one edge weight changes by 1 (in reality their definition is slightly more general).

Overview of results:
They provide an algorithm that, under the approximate-differential privacy setting, can release all n^2 shortest paths simultaneously up to error roughly \tilde{O}(sqrt{n}), while preserving privacy between adjacent datasets. This improves over the previous best work of Sealfon (PODS, 2016), which could do \tilde{O}(n) error, or \tilde{O}(sqrt{n}) error if the weights were sufficiently small (such as at most polylog(n)). They also consider special graphs where the underlying graph topology (i.e., the unweighted part of the graph) has feedback vertex set number k, meaning there exist k vertices that if you remove them you are left with no cycles. In this setting, they show one can get O(k*polylog n) error. Finally, they also provide some rudimentary experiments showing that on synthetic graphs, the error is in fact even smaller.

**Questions:**

N/A

**Limitations:**

This paper is almost entirely theoretical. While they do not discuss any potential negative societal impact, I cannot think of anything that needs to be said here so I don't think discussing this is necessary.

**Strengths And Weaknesses:**

1) The main strength is that this is an interesting open question and significantly improves over a previous paper of Sealfon which was the PODS 2016 best student paper. It also resolves a main question which was also posed in the differentialprivacy.org website.
2) The algorithms are simple and seem easy to implement.
3) The technique for the O(sqrt{n}) error is based on a rather classic method of random hitting sets, meaning a random subset of vertices is well connected to the rest of the graph. While hitting sets have been used a lot in graph algorithms, using hitting sets in the correct way is still tricky, though does not seem to require incredibly clever ideas.
4) The feedback vertex set result I believe is much more straightforward, and it is not clear whether graphs have small feedback vertex number in practice. This is a minor weakness since their main result appears to be the general graph case.
5) There are some other papers which also studied this problem at the same time. These other papers also prove other results not in this work, such as proving a lower bound of n^{1/6}, and doing better than n^{1/2} when the weights are not too large. The authors indeed acknowledge these concurrent works in the appendix, and this also suggests this is an important problem if multiple groups were interested in this problem, so this is not entirely a disadvantage.

---

> ### Author Response · Authors · 2022-08-02
> **Response to Reviewer vGTs**
>
> Thanks for your valuable comments.
>
> 4. In practice, there are common example graphs with small feedback vertex number, e.g., transportation routing maps such as the New York City MTA map and the US railway map. For these graphs, the number of feedback vertices is smaller than the number of interchange stops/cross stations, which is usually small compared to the graph size. After removing these nodes, the remaining graph becomes a forest. Thus, the error can be very small according to our theory in Section 4.
>
> 5. Yes, we were aware of the two concurrent works, and we have discussed the difference between our work and theirs in the Appendix. As you kindly pointed out, this indicates the importance of this problem in the community. We are glad to see more research interests and results on this problem and have descent respect to every try in this topic. We are happy to move the citation of these two concurrent works to the main paper in revision.

---

> > ### Comment · Reviewer_vGTs · 2022-08-07
> > **Response to Rebuttal**
> >
> > Thank you very much for your response, and for describing some examples for practical graphs with small feedback vertex number.
> >
> > I believe that if space permits, it would certainly be better for the citation to independent work be moved to the main paper. Assuming your paper is accepted I believe you are given 1 extra page for the main body, so I think this should be doable.

---

> > > ### Author Response · Authors · 2022-08-08
> > > **Thank you**
> > >
> > > Dear Reviewer,
> > >
> > > Thank you for the suggestion. Very glad to hear that our rebuttal is helpful. Certainly, we will be happy to move  the content from Appendix to the main paper as you kindly suggested.
> > >
> > > Regards,
> > >
> > > Authors

---

### Official Review · Reviewer_f9J1 · 2022-07-11

**Rating:** 7
**Confidence:** 4
**Soundness:** 3 good
**Presentation:** 3 good
**Contribution:** 4 excellent

**Summary:**

The manuscript tackles the problem of estimating all pairwise shortest path distances between nodes where sensitive edge lengths have to be protected by differential privacy (while the graph topology is non-sensitive and public). The work extends prior results with sublinear error that were limited to grid graphs and trees to general graphs and thereby answers a previously open problem. Even stronger results are obtained for "nearly acyclic" graphs (removing a limited number of edges results in a forest).

**Questions:**

Q1. What is the expected performance on real-world graphs? Which topologies would work better/worse?

Q2. Suppose we limit ourselves to grid graphs and trees. How does the performance of the proposed approach compare to prior work numerically/empirically?

Q3. Is there some reason why delta is so large for Figure 1 (compared to other approximate DP works) and what is the rationale behind the other numerical values (e.g., edge weight ranges)?

**Ethics Review Area:**

["Privacy and Security (e.g., consent)"]

**Limitations:**

The performance on real-world graphs is not quite clear, e.g., which topologies would work better or not. It would be good to either declare it as an open question or to add an empirical study with 1-2 real-world graphs (where the weights ideally also relate to users). For instance, the citation graph of papers indexed in DBLP relates to users, e.g., one could use authors as nodes and suppose that the number of co-authored papers between two authors was a sensitive edge weight (representing social network type of graphs). As another example, the road network of Open Street Map is publicly available and can be cross-referenced with population grids--using the number of inhabitants along a road as sensitive edge weights (representing almost planar graphs).

**Strengths And Weaknesses:**

STRENGTHS

S1. Novelty/Significance: The work extends prior results with sublinear error that were limited to grid graphs and trees to general graphs and thereby answers a previously open problem by leveraging "canonical shortest paths" (Definition 3.4).

S2. Novelty/Significance: Even stronger results are obtained for graphs with a small feedback vertex set (minimal set of nodes whose removal turns the graph into a forest).

S3. Relevance/Presentation/Related Work/Soundness: The problem is well-motivated, the manuscript is well-written, relevant prior work is considered and the claims appear to be appropriately substantiated.

WEAKNESSES

W1. The significance would be greater with an empirical study to provide an intuition for how well the proposed techniques work on different types of real-world graphs

W2. It would be easier to replicate Figure 1 if code was provided in the supplementary material

W3. Minor issues

- l. 182 "Definition Definition 3.4" => "Definition 3.4"
- is the red/blue color for external/internal references intended?

*Edit after author rebuttal / reviewer discussion*:

The overall rating and confidence level has been increased after considering the lack of empirical evaluation in the two contemporaneous works (https://arxiv.org/pdf/2203.16476, https://arxiv.org/pdf/2204.02335) which defuses W1 a bit. Furthermore, it is presumed that the authors will follow the many suggestions by the other reviewers to improve the presentation.

---

> ### Author Response · Authors · 2022-08-02
> **Response to Reviewer  f9J1**
>
> Thanks for your valuable feedback.
>
>
> W2. We are happy to provide the simple MATLAB code if the reviewers requested. The implementation is standard and exactly follows Algorithm 1. We first build a symmetric weight matrix, then randomly sample $\sqrt{n}$ nodes to construct the shortcuts (change/add weights in place), add weight noise according to Algorithm 1, and finally compute APSP on the new weight matrix using standard functions.
>
>
> Q1. What is the expected performance on real-world graphs? Which topologies would work better/worse?
>
> Reply: We should note that, theoretically, the breakthrough of $\tilde O(\sqrt{n})$ error is based on a worst case analysis on general graphs. In general, the error would be much smaller if, for instance, the number of edges between most of the nodes is small. Two typical topologies are tree and graph with small feedback vertex number (as studied in our paper). In practice, examples include transportation routing maps, e.g., the New York City MTA map, the US railway map. For these graphs, the number of feedback vertices is smaller than the number of interchange stops/cross stations, which is usually small compared to the graph size. After removing these nodes, the remaining graph becomes a forest. Thus, the error can be very small according to our theory in Section 4.
>
>
> Q2. Suppose we limit ourselves to grid graphs and trees. How does the performance of the proposed approach compare to prior work numerically/empirically?
>
> Reply: For grid graph, the previous  error bound is $\tilde O(n^{3/4})$, we improve it to  $\tilde O(n^{1/2})$ in this paper.
>
> For trees, the proposed approach has the same error rate compared to prior work [Seafon 2016, Fan and Li 2022] as a tree and be considered as the graph with empty feedback vertex set.
>
>
> Q3. Is there some reason why delta is so large for Figure 1 (compared to other approximate DP works) and what is the rationale behind the other numerical values (e.g., edge weight ranges)?
>
> Reply3: The value of $\delta$ does not affect the error rate with respect to $n$. In fact, using smaller $\delta$ (e.g., $10^{-6}$) results in essentially same figures (growing rate in $n$). We are happy to add more figures with smaller $\delta$ to the paper for completeness. Thank you for the suggestion.
>
>
> L1. Yes, the main contribution of our paper is to improve the theoretical error rate of APSP release, where all prior works are mostly theoretical. As you mentioned, there are indeed many possible practical application of this problem, and an empirical study on specific real-world problems has not been conducted in literature. We agree that this is also an important and interesting direction. We are happy to clarify this point and call for related future works in the paper as you kindly suggested. Thank you.

---

> > ### Comment · Reviewer_f9J1 · 2022-08-07
> > **Thank you**
> >
> > Thank you for the reply, it made some things a bit clearer.

---

> > > ### Author Response · Authors · 2022-08-07
> > > **Thank you**
> > >
> > > Dear Reviewer f9J1,
> > >
> > > Thank you for your reply. We are glad to know that the rebuttal has helped.
> > >
> > > Best Regards,
> > >
> > > Authors

---

### Meta-Review · Area_Chair_nrQB · 2022-08-23

**Recommendation:** Accept
**Confidence:** Certain

**Metareview:**

The paper makes an important contribution on DP graph optimization literature. Most reviewers found the paper well written with no serious doubts regarding the correctness. We hope authors incorporate the comments from the reviewers in their final revision to improve the presentation.

**Award:**

No

---

### Decision · Program_Chairs · 2022-09-14

Accept